# Biochar and Manure Co-Application Increases Rice Yield in Low Productive Acid Soil by Increasing Soil pH, Organic Carbon, and Nutrient Retention and Availability

**DOI:** 10.3390/plants13070973

**Published:** 2024-03-28

**Authors:** Dong Liang, Yunwang Ning, Cheng Ji, Yongchun Zhang, Huashan Wu, Hongbo Ma, Jianwei Zhang, Jidong Wang

**Affiliations:** 1Scientific Observatory and Experimental Station of Arable Land Conservation of Jiangsu Province, Ministry of Agriculture and Rural Affairs, Institute of Agricultural Resources and Environment, Jiangsu Academy of Agricultural Sciences, Nanjing 210014, China; dliang@jaas.ac.cn (D.L.);; 2Key Laboratory of Saline-Alkali Soil improvement and Utilization (Coastal Saline-Alkali Lands), Ministry of Agriculture and Rural Affairs, Nanjing 210014, China

**Keywords:** rice, wood biochar remediation, low-productive soil, pig manure, soil layers, rice–soil interactions

## Abstract

In recent years, overuse of chemical fertilization has led to soil acidification and decreased rice yield productivity in southern China. Biochar and manure co-application remediation may have positive effects on rice yield and improve acid paddy soil fertility. This study was conducted to understand the effects of co-application of wood biochar and pig manure on rice yield and acid paddy soil quality (0–40 cm soil layers) in a 5-year field experiment. The experiment consisted of six treatments: no biochar and no fertilizer (CK); biochar only (BC); mineral fertilizer (N); mineral fertilizer combined with biochar (N + BC); manure (25% manure N replacing fertilizer N) combined with mineral fertilizer (MN); and manure combined with mineral fertilizer and biochar (MN + BC). Total nitrogen application for each treatment was the same at 270 kg nitrogen ha^−1^y^−1^, and 30 t ha^−1^ biochar was added to the soil only in the first year. After five years, compared with N treatments, N + BC, MN, and MN + BC treatments increased the rice yield rate to 2.8%, 4.3%, and 6.3%, respectively, by improving soil organic matter, total nitrogen, and available phosphate under a 0–40 cm soil layer. MN + BC had the strongest resistance to soil acidification among all the treatments. The interaction between fertilizers and biochar application was significant (*p* < 0.05) in rice yield, soil electrical conductivity (10–20 cm), and soil available phosphate (20–40 cm). Principal component analysis indicated that the effect of manure on soil property was stronger than that of biochar in the 0–40 cm soil layer. The overall rice yield and soil fertility decreased in the order of biochar + mineral fertilizer + manure > mineral fertilizer + manure > biochar + mineral fertilizer > mineral fertilizer > biochar > control. These results suggest that biochar and manure co-application is a long-term viable strategy for improving acid soil productivity due to its improvements in soil pH, organic carbon, nutrient retention, and availability.

## 1. Introduction

Over-cultivation and global climate change have led to severe land degradation worldwide. Salinization, acidification, and low fertility of soil degradation threaten food security [1]. As the world’s population continues to grow, the challenge is to protect the ecological environment while ensuring food security. Rice is the main food for most people in south China [2]. Acidic soils pose significant challenges to rice cultivation, particularly in regions such as Jiangsu, China, where acidification is prevalent. The paddy soils in the Jiangsu Yangtze River Delta region exhibit widespread distribution across plains and riverbanks, primarily formed from sediment deposits carried by the Yangtze River and its tributaries, providing nutrients for rice cultivation [3]. Key factors influencing soil fertility include organic matter content, soil pH value, salt, and nutrient supply. Excessive use of chemical fertilizers exacerbates soil acidity while depleting organic matter, leading to reduced soil fertility and compromised crop yields [4]. Therefore, appropriate fertilization management measures are necessary to maintain soil fertility and ecological balance.

In addressing these challenges, a promising solution lies in the combined application of biochar and manure. Manure is an important source of nutrients for replacing mineral fertilizer. Manure, derived from natural sources, offers a rich blend of nutrients vital for plant growth and soil health [5]. Manure contains alkaline substances and abundant organic matter that can enhance the structure of acidic paddy soil, raise soil pH, increase soil organic matter content, improve soil water and nutrient retention capacity, and enhance soil aeration and drainage [6]. The large number of microorganisms in manure can promote the activity of soil microbes, improve soil biological activity, and help enhance soil fertility and productivity [7].

In addition to manure, biochar is also effective in improving acidic paddy soil. Biochar, a carbon-rich material produced through the pyrolysis of biomass, has emerged as a sustainable soil amendment with the potential to enhance soil properties and promote crop productivity [8]. Biochar undergoes high-temperature pyrolysis or thermal cracking, resulting in high stability, resistance to decomposition, and the ability to persist in soil for extended periods, maintaining its effectiveness [9]. Biochar possesses abundant micropores and a high surface area, which endow it with excellent adsorption and storage capabilities, making it beneficial for adsorbing organic matter, nutrients, and water [10]. Biochar application enhances acidic paddy soil by improving soil structure, regulating pH levels, supplying essential trace elements, increasing soil organic matter, and reducing nutrient loss [11,12]. These benefits collectively promote rice growth and lead to higher yields in acidic soil conditions.

Compared to the separate application of biochar, the combined application of biochar and manure may have a greater impact on improving soil fertility and crop productivity [13,14]. The advantages of biochar over manure lie in its ability to enhance long-term soil structure, retain nutrients effectively, improve soil aeration, regulate pH levels, and provide sustained soil amelioration effects [15]. Conversely, manure excels at rapidly supplying abundant organic matter and essential nutrients to plants, promoting plant growth, enhancing soil microbial activity, and improving soil structure and water retention capacity [16]. Biochar and manure could both release dissolved organic carbon and affect plant-microbe interactions [17]. Therefore, biochar has the advantage of long-term soil structure and fertility improvement, while manure excels in providing abundant nutrients and rapidly promoting crop growth. Combining the use of biochar and manure can leverage their respective strengths to enhance soil fertility and crop yield.

In this study, we investigate the efficacy of integrating biochar and manure as a holistic approach to ameliorating acidic paddy soils and enhancing rice yields in the Jiangsu Yangtze River Delta region. Wood biochar is derived from waste materials such as pruned branches and trimmings from road and landscaping maintenance. While straw and other types of biochar can also be used to improve acidic rice paddy soil, wood biochar is generally considered more effective in this regard due to its chemical properties, stability, and long-lasting soil improvement effects [18,19]. Additionally, it can be used without seasonal change. By harnessing the synergistic effects of biochar and manure, we aim to address the multifaceted challenges associated with soil acidity while promoting sustainable rice production practices. Through comprehensive field trials and soil analyses, we seek to elucidate the mechanisms underlying the beneficial impacts of biochar and manure on soil pH regulation, nutrient retention, soil carbon sequestration, and ultimately, rice growth and yield. Our findings aim to provide valuable insights into the potential of biochar and manure as environmentally friendly and economically viable strategies for improving soil fertility and agricultural productivity in acidic rice-growing environments.

## 2. Results

### 2.1. Effects of Biochar and Manure on Rice Yield and Yield Components

Biochar and manure application significantly increased rice yield over the last three years (*p* < 0.05) (Table 1). Compared with the N treatment, the increasing rates of grain yield of the N + BC, MN, and MN + BC treatments were 2.8%, 4.3%, and 6.3%, respectively. The N and MN treatments significantly increased the panicles per plant and grains per panicle (*p* < 0.05). A negative impact on the 1000-grain weight was observed in the N treatment (*p* < 0.05). The overall rice yield decreased in the order of MN + BC > MN > N + BC > N > BC > CK. Fertilizers or biochar application significantly affected grain yield in different treatments (ANONA), and fertilizer and biochar application had a significant effect on this trait.

### 2.2. Effect of Biochar and Manure on Paddy Soil pH and Electrical Conductivity (EC)

Generally, biochar and manure addition both increased the soil pH in all the treatments after five years (Figure 1a). The pH of CK in the 0–10 cm, 10–20 cm, and 20–40 cm soil layers was 6.35, 6.52, and 6.94, respectively. In the 0–10 cm, 10–20 cm, and 20–40 cm soil layers, the pH values of each treatment were increased or decreased by BC (+0.21, +0.16, and +0.12 units), N (−0.76, −0.34, and −0.08 units), N + BC (−0.49, −0.15, and +0.04 units), MN (−0.41, +0.06, and +0.04 units), and MN + BC (−0.22, +0.18, and +0.13 units) compared with CK. The soil acidification in the upper layer was worse than that in the underlying layer. The addition of biochar to the BC and N + BC treatments significantly improved the pH (*p* < 0.05) in all the soil layers compared with those of the CK and N treatments, respectively. The manure application in the MN treatment significantly improved the pH (*p* < 0.05) in all the soil layers compared with those in the N treatment. The MN + BC treatment improved the pH significantly (*p* < 0.05) in only the 0–10 cm layer compared with those in the MN treatment. Fertilizers or biochar application significantly affected pH in different treatments and soil layers (ANONA), but the interaction between fertilizers and biochar application was not significant.

However, soil EC was decreased by biochar, but increased by manure. Biochar addition significantly decreased the soil EC in both the CK + BC (all soil layers) and N + BC (0–10 and 10–20 cm soil layers) treatments (*p* < 0.05) after five years (Figure 1b). In addition, biochar addition significantly decreased the soil EC of MN + BC (*p* < 0.05) only in the 0–10 cm soil layer compared with that of MN. The soil EC in the upper layer was lower than that in the underlying layer. In the 0–10 cm soil layer, the overall order of EC was MN > CK > BC > N (*p* < 0.05). In both the 10–20 cm and 20–40 cm soil layers, the EC of all the treatments followed the same order: MN > N > CK > BC. In the 20–40 cm soil layer, there was no significant effect (*p* > 0.05) of biochar addition in either the N + BC or MN + BC treatments. Fertilizers or biochar application significantly (*p* < 0.0001) affected soil EC in different treatments and soil layers (ANONA), but the interaction between fertilizers and biochar application was only significant (*p* < 0.0001) in the 10–20 cm soil layer.

### 2.3. Effect of Biochar and Manure on Field Moisture Capacity (FMC)

Generally, biochar increased FMC, but manure decreased it. Biochar addition increased FMC in all the treatments after five years (Figure 2). The FMC of CK in the 0–10 cm, 10–20 cm, and 20–40 cm soil layers was 30.1%, 25.3%, and 23.4%, respectively. In the 0–10 cm, 10–20 cm, and 20–40 cm soil layers, the FMC values of each treatment were increased or decreased by BC (+2.40, +1.37, and +0.83 units), N (+3.10, −3.68, and −0.90 units), N + BC (+ 4.91, −1.96, and −0.33 units), MN (−2.83, −4.90, and −3.40 units), and MN + BC (−1.32, −3.10, and −1.48 units) compared with CK. In the 0–10 cm soil layer, the overall order of FMC was N + BC > N > BC > CK > MN + BC > MN. In both the 10–20 cm and 20–40 cm soil layers, the FMC of all the treatments followed the same order: CK and BC > N and N + BC > MN and MN + BC. The manure application in the MN treatment significantly decreased the FMC (*p* < 0.05) in all the soil layers compared with those of the CK and N treatments. The N treatment improved the FMC significantly (*p* < 0.05) only in the 0–10 cm layer compared with those with the CK treatment. Fertilizers or biochar application significantly (*p* < 0.001) affected FMC in different treatments and soil layers (ANONA), but the interaction between fertilizers and biochar application was not significant.

### 2.4. Effect of Biochar and Manure on Soil Organic Matter (SOM)

Generally, biochar and manure addition both significantly increased SOM in all the treatments and soil layers (*p* < 0.05) after five years (Figure 3). The SOM contents of CK in the 0–10, 10–20, and 20–40 cm soil layers were 26.3, 14.0, and 13.1 g kg^−1^, respectively. In the 0–10 cm, 10–20 cm, and 20–40 cm soil layers, the SOM values of each treatment were increased or decreased by BC (+5.3, +1.2, and +1.0 units), N (+6.4, +1.1, and −0.7 units), N + BC (+9.4, +2.0, and +0.9 units), MN (+8.1, +2.3, and +4.1 units), and MN + BC (+10.4, +3.0, and +5.1 units) compared with CK. In the 0–10 and 10–20 cm soil layers, the overall order of SOM was MN > BC and N > CK (*p* < 0.05). In the 20–40 cm soil layer, fertilization with N did not significantly improve SOM (*p* < 0.05). The application of biochar or manure increased SOM significantly (*p* < 0.05) in all the soil layers. Fertilizers or biochar application significantly (*p* < 0.0001) affected SOM in different treatments and soil layers (ANONA), but the interaction between fertilizers and biochar application was not significant.

### 2.5. Effect of Biochar and Manure on Paddy Soil Total Nitrogen, Available Phosphorus, and Available Potassium

Generally, biochar addition increased soil total nitrogen at the 0–10 cm (BC and N + BC treatments) and 10–20 cm (BC and MN + BC treatments) soil layers after five years (Figure 4a). Total nitrogen decreased with the soil depth, such that the level of total nitrogen in the CK treatment at the 0–10, 10–20, and 20–40 cm soil layers was 1.51, 0.93, and 1.04 g kg^−1^, respectively. In the 0–10 cm soil layer, the overall increase in the level of total nitrogen in the treatments compared with that of CK followed the order N + BC (38.1%) > N, MN, and MN + BC (26.0%–28.4%) > BC (7.3%) > CK (*p* < 0.05); in the 10–20 cm soil layer, the order of total nitrogen increase was N + BC and N (24.4–27.6%) > MN + BC (19.1%) > MN (12.5%) > BC (6.1%) > CK (*p* < 0.05). In the 20–40 cm soil layer, the order of total nitrogen increase was N, N + BC, MN, and MN + BC (3.5–12.2%) > BC and CK (*p* < 0.05). Fertilizers (all soil layers) or biochar (except for the 20–40 cm soil layer) application significantly (*p* < 0.05) affected total nitrogen in the different treatments (ANONA), but the interaction between fertilizers and biochar application was not significant.

Generally, biochar addition increased the soil available phosphorus significantly (*p* < 0.05) in the BC and N + BC treatments compared with those in the CK and N treatments in all the soil layers after five years (Figure 4b). However, biochar application in the MN + BC treatment only decreased available phosphorus significantly (*p* < 0.05) compared with MN in the 20–40 cm soil layer. AP decreased with the soil depth, such that the available phosphorus of CK at the 0–10 cm, 10–20 cm, and 20–40 cm soil layers was 6.3, 3.1, and 1.5 g kg ^−1^, respectively. In the 0–10 cm soil layer, the overall increase in the level of available phosphorus in the treatments compared with that of CK followed the order MN + BC and MN (127–130%) > N + BC (78%) > N (63%) > BC (17%) > CK (*p* < 0.05); in the 10–20 cm soil layer, the order of available phosphorus increase was MN + BC and MN (32–44%) > BC (22%) > N + BC and CK (0–6.4%) > N (−15%) (*p* < 0.05). In the 20–40 cm soil layer, the order of available phosphorus increase was N + BC and MN (43–50%) > MN and BC (35–41%) > N (27%) > CK (*p* < 0.05). Fertilizers or biochar application significantly (*p* < 0.05) affected available phosphorus in different treatments (ANONA), but the interaction between fertilizers and biochar application was only significant in the 20–40 cm soil layer (*p* < 0.0001).

Generally, biochar and manure addition both increased the soil available potassium significantly (*p* < 0.05) after five years (Figure 4c). However, fertilizer and manure application in the N and MN treatments significantly decreased available potassium (*p* < 0.05) compared with that in the CK treatment. Available potassium increased with the soil depth; for example, the available potassium of CK in the 0–10, 10–20, and 20–40 cm soil layers was 102, 110, and 119 g kg^−1^, respectively. In the 0–10 cm soil layer, the overall increase in the available potassium level compared with that of the CK treatment was BC (+15.7%) > CK > MN + BC (+9.5%) > MN (+3.1%) > N + BC (−11.7%) > N (−20.8%) (*p* < 0.05). In the 10–20 cm soil layer, the order of available potassium increase was BC (+7.3%) > CK > MN + BC (+0.3%) > MN (−2.7%) > N + BC (−4.2%) > N (−8.5%) (*p* < 0.05). In the 20–40 cm soil layer, the order of available potassium was BC (+3.4%) > CK > MN + BC (+4.4%) > N + BC (+3.7%) > MN (−5.0%) > N (−11.9%) (*p* < 0.05). Fertilizers or biochar application significantly (*p* < 0.01) affected available potassium in different treatments (ANONA), but the interaction between fertilizers and biochar application was not significant.

### 2.6. Correlation Analysis of Soil Property and Rice Yield

Pearson’s correlation was applied to analyse the relationship between soil properties in different soil layers and paddy yield. Figure 5 shows that rice yield had a close correlation with soil properties. In the 0–10 cm soil layer, soil organic matter, total nitrogen, and available phosphorus had significant positive correlations with rice yield (*p* < 0.01), but pH and available potassium had significant negative correlations with rice yield (*p* < 0.05). In the 10–20 cm soil layer, soil organic matter, total nitrogen, and electrical conductivity had significant positive correlations with rice yield (*p* < 0.05), but field moisture capacity and available potassium had significant negative correlations with rice yield (*p* < 0.05). In the 20–40 cm soil layer, electrical conductivity, soil organic matter, total nitrogen, and available phosphorus had significant positive correlations with rice yield (*p* < 0.05), but field moisture capacity and available potassium had significant negative correlations with rice yield (*p* < 0.01).

### 2.7. PCA of the Soil Properties

PCA was applied to evaluate the differences in the soil properties, and it treated the following seven soil properties of each soil sample as one matrix: pH, electrical conductivity, field moisture capacity, soil organic matter, total nitrogen, available phosphorus, and available potassium. Figure 6 shows the first and second principal components (PC1 and PC2) of the seven soil properties at the 0–10 cm (92.2% total variance), 10–20 cm (98.7% total variance), and 20–40 cm (99.1% total variance) soil layers. The longer the distance between samples, the more dissimilar they are in terms of soil properties. In the 0–10 cm soil layers, the N and N + BC samples were similar to each other, which indicated their soil properties being similar compared with those of other treatments. The MN and MN + BC samples also had similar soil properties. The effects of mineral fertilizer and manure application on soil properties were stronger than those of biochar in the 0–10 cm soil layer. In the 10–20 cm soil layers, the BC, N + BC, MN, and MN + BC samples were similar to each other. This result indicated that the effect of biochar application on soil properties was stronger than that of fertilizer application. In the 20–40 cm soil layers, the MN and MN + BC samples overlapped with each other, which shows that the effect of biochar application combined with manure was not significant in this soil layer.

## 3. Discussion

### 3.1. Co-Application of Biochar and Manure Improve Rice Productivity and Soil Fertility

Our results are consistent with the report [20] showing that applying manure and biochar separately or in combination significantly increased crop yields. When compared to separate applications, the application of biochar and manure combined produced the highest rice production by increasing soil pH, organic matter, total nitrogen, and available phosphorus. The suitable soil pH and available nutrient content affect rice root growth and nutrient uptake, which typically supports higher rice yields [21]. Co-application of biochar and manure increased soil pH, reduced the accumulation of soil acidity, and improved the soil environment of acidic paddy soil. The appropriate soil pH promotes the growth and activity of soil microorganisms, which participate in nutrient cycling and organic matter decomposition, exerting a significant influence on rice growth [22]. Both biochar and manure contain abundant organic matter. Biochar’s stability and manure’s richness synergize to enrich soil organic matter, stabilise carbon, and enhance soil structure, collectively fostering soil carbon sequestration [23,24]. Studies [25,26] suggest that higher soil carbon content contributes to improved soil structure, enhanced soil fertility, and beneficial microbial growth, collectively enhancing rice growth conditions and yield. Biochar may enhance nutrient fixation and release rates, thus improving nutrient effectiveness in the soil [9,12]. Meanwhile, manure stimulates soil microbial activity and further promotes nutrient release and transformation [7]. The combined application of biochar and manure maximises their advantages, collectively improving nutrient utilization efficiency in the paddy soil and providing a more favourable soil environment for rice growth.

### 3.2. Co-Application of Biochar and Manure Improve Soil Carbon Storage

Both biochar and manure are rich in organic carbon, increasing soil organic matter content and the capacity of the soil carbon pool. When biochar is ageing, the interaction between biochar and soil continues to occur, such as through physical weathering and biochemical reactions [27], and the ageing process of biochar causes it to disintegrate into small particles [28]. We found that the wood biochar increased soil organic carbon in both the upper and underlying soil layers over five years. The aged biochar particles may transform into deeper soil layers. It was reported that the addition of straw biochar to the paddy soil increased the soil organic carbon only in the first year, but a few years later, the soil organic carbon content was in a downward trend [11,29]. The wood biochar may be more stable than the straw biochar. On the one hand, biochar has excellent carbon sequestration characteristics, reducing the rate of organic matter decomposition, decreasing carbon and nitrogen losses, and enhancing soil carbon stability and persistence [30]. On the other hand, manure provides a favourable growth environment and carbon source for soil microbes, promoting microbial activity and increasing microbial diversity and population, thereby promoting organic matter decomposition and soil carbon accumulation [31]. The application of manure can increase soil aggregate stability and enhance the sequestration of organic carbon and nitrogen [32]. Manure additions can change the amount of various organic carbon functional groups in the soil by raising the amount of carboxyl carbon, which can impact soil carbon cycling and stability [33]. Furthermore, manure can alter the distribution and content of various organic carbon fractions in the soil, which can have an impact on the soil’s ability to maintain fertility [34]. Both biochar and manure can also alleviate soil acidification, increase soil pH, and facilitate the stable accumulation of organic matter. Therefore, the combined use of biochar and manure can synergistically enhance soil organic carbon stability and availability, promote soil microbial activity, and consequently increase crop yields.

### 3.3. Co-Application of Biochar and Manure Improve Soil Nutrient Retention and Availability

On the one hand, biochar can stimulate the activity of enzymes such as urease and peroxidase, which promote the growth and activity of beneficial bacteria in the soil; in addition, biochar boosts microbial diversity and abundance and enhances the structure of the soil microbial community. By affecting microbial metabolic activities, which include the cycling and transformation processes of elements such as carbon, nitrogen, and phosphorus, biochar has the potential to affect crop development and soil nutrient availability [35]. The effects of biochar on soil respiration are complex, and the type and characteristics of the material can have varying effects on the activity of soil microbes that respire. Biochar can indirectly promote plant growth and development and increase crop yield by improving soil environment and microbial communities [36]. On the other hand, the organic matter in manure can provide energy and nutrients for soil microorganisms, promoting the release and transformation of nutrients in the soil. This facilitates the enhancement of plant nutrient absorption and utilization [32]. Long-term application of manure significantly increases the organic carbon and total nitrogen content in the soil, leading to the accumulation of soil nutrients. The addition of manure also enhances the stability of nutrients in the soil, reducing nutrient leaching and runoff, thereby improving the nutrient utilization efficiency of the soil [34]. Therefore, the co-application of biochar and manure can result in synergistic enhancement of soil nutrient utilization efficiency.

The combined application of biochar and manure did not increase soil nitrogen concentration but resulted in the highest yield, indicating high nitrogen utilization efficiency. The mechanism by which the combined application of biochar and manure improves soil nutrient retention and availability involves various biological, chemical, and physical processes. Nitrogen uptake is always low in the paddy field because of ammonia volatilization and nitrogen leaching in the field [37]. Biochar’s highly porous structure allows it to adsorb nitrogen, reducing nitrogen leaching and volatilization, thereby improving nitrogen utilization efficiency [38]. This helps minimise nitrogen losses and keeps more nitrogen available in the soil for plant uptake. Organic nitrogen in manure can gradually decompose and be released over time into inorganic nitrogen forms, such as ammonium and nitrate, which can be converted into plant-available forms through microbial decomposition [39].

The combined application of biochar and manure can enhance phosphorus availability and uptake by plants through mechanisms such as adsorption, pH modification, organic matter decomposition, soil structure improvement, microbial activity enhancement, and reduction of phosphorus fixation [13]. Biochar has a high phosphorus adsorption capacity, which can reduce phosphorus loss [40]. In acidic soils, phosphorus can become fixed by aluminium and iron oxides, rendering it unavailable to plants. Biochar can bind with these metal ions, reducing their availability for phosphorus fixation and improving phosphorus availability for plants [41]. Manure contains organic phosphorus, which can be converted into inorganic phosphorus through microbial action, enhancing phosphorus availability [42]. As soil pH increases, the solubility and availability of phosphorus also increase, making it more accessible to plants for uptake [15].

Potassium is an important participant in the physiological process of rice. Biochar contains a considerable amount of potassium and can serve as a potassium supplement. Biochar possesses adsorption capacity and may adsorb a portion of potassium ions in the soil, although its release rate may be relatively slow [43]. The addition of biochar can enhance soil structure by increasing soil aggregates and porosity, as well as increasing water retention capacity [44]. This benefits root growth and potassium uptake by plants [45]. The organic matter in manure provides carbon sources and energy for soil microorganisms, promoting the growth and activity of soil microorganisms. Some microorganisms release potassium elements during the decomposition of organic matter, increasing the available potassium content in the soil for crops [46]. When the pH of acidic soil is low, potassium elements in the soil are easily fixed and difficult for plants to absorb and utilize [47]. By applying biochar and manure, soil pH can be increased, promoting the release of potassium elements in the soil, and facilitating their absorption by plants.

In summary, the combined application of biochar and manure on the effectiveness of nitrogen, phosphorus, and potassium nutrients in acidic paddy soil involves the synergistic effects of biochar’s ion exchange capacity for nitrogen, phosphorus, and potassium, enhanced microbial activity facilitating nutrient transformation, the decomposition of organic matter releasing nutrients to improve nutrient utilization efficiency, and soil structure improvement increasing nutrient retention capacity.

### 3.4. Interaction Effect of Biochar and Manure on the Soil–Rice System

The co-application of biochar and manure increased rice production. By supplying nutrients and enhancing the soil properties, biochar and manure co-application may increase rice’s nutrient use efficiency. Biochar can enhance nutrient retention and reduce leaching [38], while manure provides readily available nutrients [42]. Together, they can improve nutrient cycling and availability, promoting healthier rice growth. Biochar’s alkaline properties can help mitigate soil acidity, while manure decomposition can also contribute to pH changes [48]. The combined application can create a more favourable environment for rice growth. Biochar improves soil structure by increasing aggregation and porosity, enhancing soil water holding capacity, while manure adds organic matter. This combined effect improves soil aeration, water infiltration, and root development, ultimately benefitting rice growth and yield. Both biochar and manure support microbial activity in the soil. Microorganisms play a crucial role in nutrient cycling and decomposition processes. The combination of biochar and manure can stimulate microbial populations, enhancing nutrient mineralisation and availability for rice plants [46]. The improved soil fertility, nutrient availability, organic carbon, and enhanced soil–water relationships contribute to better crop performance and higher yields.

## 4. Materials and Methods

### 4.1. Site Description

The experimental site is located in Li Yang City, Jiangsu Province, China (31°29′17″ N, 119°19′57″ E), which is a subtropical monsoon climate and had a mean rainfall of 1149 mm and a mean annual temperature of 15.4 °C throughout 2000–2020. A rotation system of summer rice followed by winter wheat or rape seed has been used for decades in this area. The soil is acidic (pH = 6.4). The particle size distribution was 21% sand, 43% silt, and 36% clay. The soil bulk density is 1.05. The soil aggregates of large macroaggregates (≥2 mm), small macroaggregates (2–0.25 mm), microaggregates (0.25–0.053 mm), and silt and clay-sized particles (≤0.053 mm) are 2.4%, 11.6%, 32.2%, and 46.2%, respectively.

It was a paddy soil (the year 2016) with a soil organic matter content of 35.7 g kg^−1^ and a soil total nitrogen content of 1.56 g kg^−1^. The available phosphate and available potassium contents were 18.6 mg kg^−1^ and 134 mg kg^−1^, respectively. The paddy soil is classified as a hydroagric Stagnic Anthrosol (Chinese Soil Taxonomy) or Entic Halpudept (USDA Soil Taxonomy).

### 4.2. Biochar, Fertilization, and Manure Properties

Biochar was provided by San Yang Energy Company (Jiangsu Province, China), and produced from camphor tree wood at a temperature range of 450–550 °C. Biochar was alkaline with pH 9.43, and 68.5% organic carbon, 0.46% N, 1.3% P and 2.4% K. The total amounts of Ca, Mg, Si, and Fe were 0.8%, 0.6%, 3.1%, and 0.3%, respectively.

In the experiment, urea, calcium phosphate, and potassium chloride were used as fertilizers for nitrogen, phosphate, and potassium, respectively. The pig compost was used as the manure, which contained approximately 350 g kg^−1^ organic carbon, 20.3 g kg^−1^ N, 42.1 g kg^−1^ P_2_O_5_, and 23.4 g kg^−1^ K_2_O on a dry weight basis.

### 4.3. Experimental Design

The experiment began in May 2016 and ended in November 2021. The experiment is based on a two-factor experimental design. The first factor is wood biochar amendment, and the other is N fertilization and manure. Biochar was amended only one time at the rates of 0 and 30 t ha^−1^ in 2016. Nitrogen fertilizer was applied at rates of 270 kg ha^−1^ (fertilizer N only) and 270 t ha^−1^ (25% manure N replacing fertilizer N) every year. There were 6 treatments in total, including the following: (1) no fertilizer treatment (control, CK); (2) biochar treatment (30 t ha^−1^ in 2016, BC); (3) N chemical fertilizer (270 kg N ha^−1^y^−1^, N); (4) N fertilizer combined with biochar (N + BC); (5) manure (containing 67.5 kg N ha^−1^y^−1^) combined with N fertilizer (containing 202.5 kg N ha^−1^y^−1^) (MN); and (6) manure combined with N fertilizer and biochar (MN + BC). Each treatment was replicated 3 times, which yielded 18 experimental plots. Each experimental plot covered an area of 20 m^2^ (4 m × 5 m). The rice variety Nangeng 46 (*Oryza sativa* L.) was transplanted by hand in mid-June with a plant spacing of 30.0 cm × 13.5 cm. Crop management was carried out following the farming practices.

The field experiment was set up in May 2016. Before application, the biochar was sieved through a 2-mm sieve and homogenised. Biochar was broadcast onto the soil surface first and then incorporated into the top layer by ploughing to a depth of 5 cm. No biochar was added in the following years. Nitrogen fertilizer was applied every rice season throughout the 5-year experiment. For the rice season, 30% of the N fertilizer was used as base fertilizer before rice transplanting, 40% as top dressing at the tillering stage, and the remaining 30% at the panicle stage. Both P and K fertilizers were applied as base fertilizers at rates of 60 kg P_2_O_5_ ha^−1^y^−1^ and 96 kg K_2_O_5_ ha^−1^y^−1^. Manure was also applied as a base fertilizer. The nutrient index of manure was measured before application. According to the principle of total nutrient consistency, the phosphorus and potassium deficiencies of the manure were supplemented by fertilizer in the MN and MN + BC treatments. The annual application of total N remained the same in all the treatments. Rice was flooded in July and August and drained in September. Rice was harvested at the end of October or the beginning of November. The fallow tillage time was from December to April. Every season, the rice straw was removed from the field.

### 4.4. Sampling and Measurements

Rice grain samples were collected and analysed in the years 2019–2021. In each treatment at the rice maturity stage, a 5-m^2^ area in the centre of each plot was harvested to determine grain yield at the standard moisture content of 13.5%.

Soil samples were taken from the 0–10 cm, 10–20 cm, and 20–40 cm soil layers of each plot after rice harvest in 2021. The fresh soil samples were transported to the laboratory, homogenised manually, and air dried. The samples were sieved to <0.15 mm. Soil pH (pH meter FE28, METTLER TOLEDO, Shanghai, China) and soil electrical conductivity (EC) (EC meter 307A, LEICI, China) were measured in a 1:5 (*w*:*v*) soil-deionised water suspension. Field moisture capacity (FMC) was determined by weight loss after drying at 105 °C for longer than 24 h. SOM was measured by K_2_Cr_2_O_7_–H_2_SO_4_ oxidation. Total nitrogen was measured by the semimicro-Kjeldahl method. Available phosphorus was measured by treatment with 0.5 mol L^−1^ NaHCO_3_ (pH 8.5), followed by molybdenum blue colorimetry. Available potassium was measured by 1 mol L^−1^ NH_4_OAc extraction-flame photometry.

### 4.5. Data Processing and Statistical Analysis

All results are shown as the means of three replicates with standard deviations. One-way analysis of variance (ANOVA) was conducted to assess the effect of biochar and fertilizer. The difference between the treatments was determined by Tukey’s HSD test at the 0.05 level. The different lowercase letters represent significant differences (*p* < 0.05) between the treatments. Pearson correlation was applied to analyse the correlations between the rice yield and soil properties at the *p* < 0.05 level. Data analyses were run using SPSS version 22.0.

Principal component analysis (PCA) is a dimensionality reduction chemometrics technique that reduces redundant information in a data set. In this experiment, PCA was used to find the similarities and distinctions among the soil properties of the 0–10 cm, 10–20 cm, and 20–40 cm soil layers. PCA treated the pH, electrical conductivity, field moisture capacity, soil organic matter, total nitrogen, available phosphorus, and available potassium of each soil sample as vectors and formed linear combinations of the vectors by assigning a weight to each vector. MATLAB R2016a software was used for data analysis.

## 5. Conclusions

The results showed that co-application of biochar and manure amendment to an acidic paddy soil had a positive impact on rice yield and soil quality in the 0–40 cm soil layers. Biochar and manure co-application increased rice yield by improving soil organic carbon and available phosphorus in the 0–40 cm soil layers. Biochar and manure additions can significantly increase resistance to soil acidification, decrease soil salinity, and improve soil water capacity. The soil fertility indices of soil organic matter, total nitrogen, available phosphorus, and available potassium were increased with biochar application. In addition, the interaction between fertilizers and biochar application had a significant effect on soil electrical conductivity (10–20 cm), soil available phosphorus (20–40 cm), and rice yield. According to the PCA, the effect of manure application on soil properties was stronger than that of biochar application in the 0–40 cm soil layer. The results of this 5-year field experiment show that biochar and manure co-application can improve acidic soil conditions in the 0–40 cm soil layers as well as increase rice yield. The co-application may facilitate microbial activity and improve the root growth and rhizosphere extension. Future research should investigate the mechanisms underlying the co-application of biochar and manure on the rhizosphere extension and soil microbial community structure. In conclusion, the synergistic effect of biochar and manure in the acidic soil–rice system involves a complex interplay of factors that collectively improve soil fertility, nutrient availability, soil structure, microbial activity, and ultimately, rice yield. Further research is needed to understand the specific mechanisms underlying this interaction and to optimize the application rates and methods for sustainable rice production in acidic soil environments.

## Figures and Tables

**Figure 1 plants-13-00973-f001:**
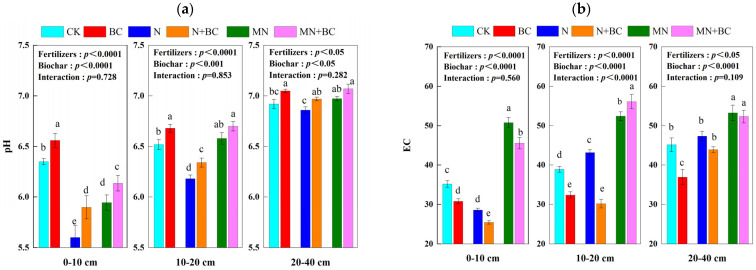
pH (**a**) and electrical conductivity (EC) (**b**) of the 0—40 cm soil layer under various treatments. Different letters in the same column indicate significant differences between the treatments (*p* < 0.05).

**Figure 2 plants-13-00973-f002:**
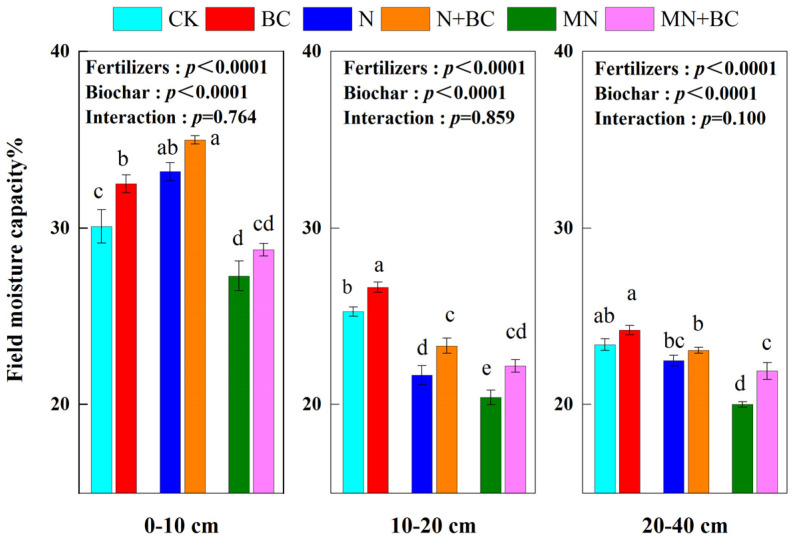
Field moisture capacity (FMC) of the 0—40 cmsoil layer under various treatments. Different letters in the same column indicate significant differences between the treatments (*p* < 0.05).

**Figure 3 plants-13-00973-f003:**
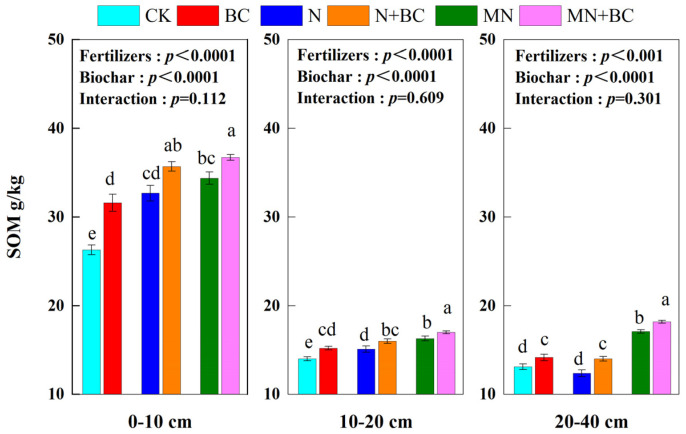
Soil organic matter (SOM) of the 0—40 cm soil layer under various treatments. Different letters in the same column indicate significant differences between the treatments (*p* < 0.05).

**Figure 4 plants-13-00973-f004:**
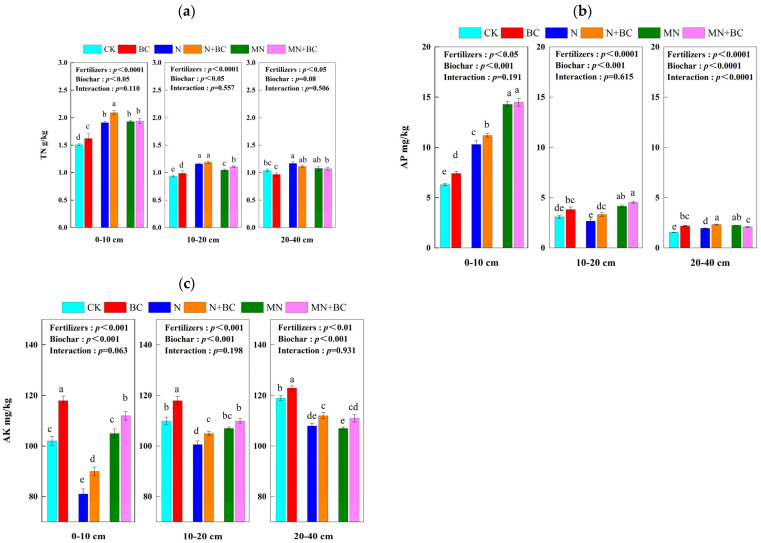
Total nitrogen-TN (**a**), available phosphorus-AP (**b**), and available potassium-AK (**c**) of the 0–40 cm soil layer under various treatments. Different letters in the same column indicate significant differences between the treatments (*p* < 0.05).

**Figure 5 plants-13-00973-f005:**
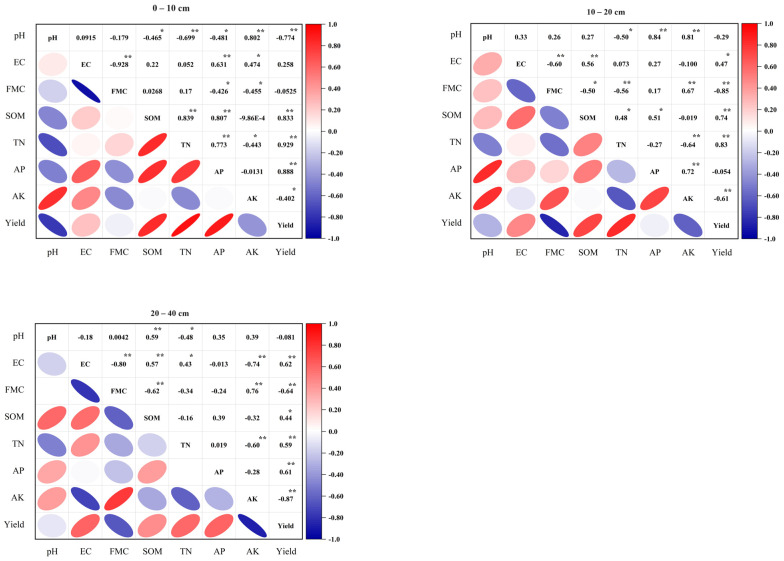
Correlation analysis between soil properties (0–40 cm soil layer) and rice yield. Note: EC: electrical conductivity; FMC: field moisture capacity; SOM: soil organic matter; TN: total nitrogen; AP: available phosphorus; AK: available potassium. * and ** indicate significant differences between the treatments (*p* < 0.05 and *p* < 0.01).

**Figure 6 plants-13-00973-f006:**
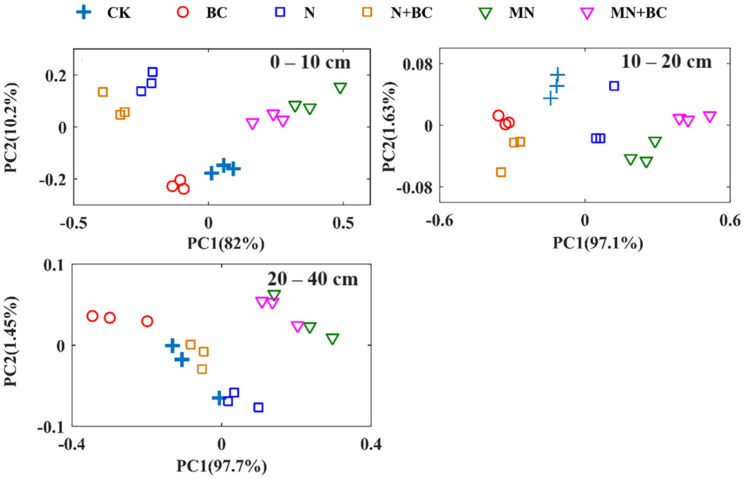
Principal component scatter plots of the soil properties (0–40 cm soil layer) for various treatments.

**Table 1 plants-13-00973-t001:** Rice yield and yield components under different treatments over the last 3 years.

Treatment	Panicles Per Plant	Grains Per Panicle	Thousand-Grain Weight (g)	Grain Yield (t ha^−1^)
CK	12.2 c	97.4 c	28.4 a	4.88 e
BC	12.1 c	109 b	27.9 ab	6.40 d
N	16.8 b	120 ab	27.2 b	8.85 c
N + BC	18.1 a	121 ab	27.1 b	9.10 b
MN	17.9 a	118 ab	27.5 ab	9.23 ab
MN + BC	17.7 ab	123 a	27.6 ab	9.41 a
Fertilizers	**	**	*	**
Biochar	*p* = 0.224	*p* = 0.061	*p* = 0.279	**
Fertilizers × Biochar	*p* = 0.058	*p* = 0.163	*p* = 0.342	*

Note: CK: the control, with no biochar and fertilizer; BC: with biochar only; N: with fertilizer only; N + BC: with fertilizer and biochar; MN: with manure and fertilizer; MN + BC: with manure, fertilizer and biochar. The yield represents the 3-year average yield from 2019 to 2021; * and ** indicate significant differences between the treatments (*p* < 0.05 and *p* < 0.01); Different letters in the same column indicate significant differences between the treatments (*p* < 0.05).

## Data Availability

Data are contained within the article.

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
