# Peer review of "Biochar and Manure Co-Application Increases Rice Yield in Low Productive Acid Soil by Increasing Soil pH, Organic Carbon, and Nutrient Retention and Availability"

_plants, 2024, doi:10.3390/plants13070973_

Round 1

Reviewer 1 Report

Comments and Suggestions for Authors

The assigned manuscript has been reviewed. The authors detailed the findings on changes in soil properties caused by integrated application of biochar and manure replacing synthetic fertilizer N. The data obtained is enormous and significant, however several issues are required to be amended. 

Title; should be revised based on contents of the manuscript, it is not clear which soil property they talked about

Abstract; Line 12, which fertilization, chemical?

A concluding statement taking into account the implications should be provided at the end to make it more engaging. 

Introduction;

The main problem in introduction is lack of focus. Only a very naive information is provided with no background of research. I strongly suggest to amend it and provide a clear flow of information, a well developed hypothesis following a gap statement. 

How are the paddy soils distributed in Jiangsu and what are the key factors affecting their fertility. This should be taken as the major engaging statement while describing their need for the study. 

What changes in soil characters are referred to as soil quality ? Lines 63-64? Did they measured any soil quality index?

M&M, 

It is essential to provide the nutrient profiles of the pig manure applied. 

The duration of experiment? How is the rates and amounts of each nutrient coming from manure and fertilizers? This is a lack of information here. 

headings 4.1 and 4.2 are the same. Rephrase them. 

Did the authors managed crop residues ? It is obvious they only managed N in all treatments (manure and or chemical ones), how about the P and K, were they considered? 

Results

The data in table 1 is average of 5 years or what?

combine figures 1 and 2. so do for 5, 6 and 7.

Discussion

In discussion, instead of repeating results such as lines 230-235, interpret them with recent published studies, 

A strong justification of how manure and biochar especially their combination can improve crop yields and soil properties. 

You need to search the literature on role of biochar and manure for crop yield enhancement and changes in soil properties and cite them into discussion. It is obvious biochar and manure brings more C into soil, this C in soil rejuvinate soil structure and other soil properties which is a key mechanisms. Also, combined action of biochar and manure brings more crop yields, more crop yields brings more straw return and a\hence more C ultimately resulting in improved crop-soil system. This aspect should be mentioned in discussion and take help of mentioned articles below for discussion as well as introduction. 

Soil aggregation and soil aggregate stability regulate organic carbon and nitrogen storage in a red soil of southern China

Long-term fertilization enhanced carbon mineralization and maize biomass through physical protection of organic carbon in fractions under continuous maize cropping

Long-term fertilization alters chemical composition and stability of aggregate-associated organic carbon in a Chinese red soil: Evidence from aggregate fractionation, C …

Food and agricultural wastes-derived biochars in combination with mineral fertilizer as sustainable soil amendments to enhance soil microbiological activity, nutrient cycling …

Comparison of the Responses of Soil Enzymes, Microbial Respiration and Plant Growth Characteristics under the Application of Agricultural and Food Waste-Derived Biochars

The mechanisms of manure and biochar are widely discussed and reported in these suggestions which might help you make discussion more better. 

Comments on the Quality of English Language

English quality is not upto the mark. There are some redundant sentences and syntax error. 

Author Response

Dear Reviewer:

Thank you for reviewing our manuscript, and your comment is very professional and constructive. Based on your comment, we have made modifications to the original manuscript. Here, we responded to your good comments point by point, as below. Please see the attachment.

Reviewer 2 Report

Comments and Suggestions for Authors

Include the Taxonomic classification of the soils used in this experiment.

Include some information on the physical properties of the soil

In the Abstract, include a statement of conclusion that shows the practical impact of your study on the use of the information by the growers.

Author Response

Dear Reviewer:

Thank you for reviewing our manuscript, and your comment is very professional and constructive. Based on your comment, we have made modifications to the original manuscript. Here, we responded to your good comments point by point, as below.

Reviewer 3 Report

Comments and Suggestions for Authors

The manuscript assess the possibility of increasing rice yields in low pH soils with biochar, manure, mineral fertilizer and their combinations. The idea of using biochar amendment to improve soil physical properties is well established, but limited work has been done to ascertain biochar impact on soil chemical properties. Authors conclude improved yield due to biochar and must show how the yield improvements relates to the soil physical properties as well.

Introduction section needs to be rewritten and latest scientific papers on rice + biochar + manure included.

The authors should clarify why they used wood based biochar in the study and not other biochar.

Several unnecessary abbreviations are used and makes it difficult to read the paper. For instance AK, AP, TN, should be written in full.

The NPK results should be presented together and not separately. In addition, the NPK results can be presented in a table for better visualization (suggestion).

The discussion section must be improved and several statements need to be referenced properly.

Comments on the Quality of English Language

An expert should review the English language for this manuscript. Errors in vocabulary, tense, and word selection need correction.

Author Response

(The authors gave the same response as above.)

Round 2

Reviewer 1 Report

Comments and Suggestions for Authors

While the authors have revised the manuscript. There are still some changes which are not revised. The authors respond that they have revised the discussion, but where they have revised it? I cannot find that. Especially for discussion section, just mentioning and repeating their results is not sufficient. Therefore, they need to work on this comment as below. 

You need to search the literature on role of biochar and manure for crop yield enhancement and changes in soil properties and cite them into discussion. It is obvious biochar and manure brings more C into soil, this C in soil rejuvinate soil structure and other soil properties which is a key mechanisms. Also, combined action of biochar and manure brings more crop yields, more crop yields brings more straw return and a\hence more C ultimately resulting in improved crop-soil system. This aspect should be mentioned in discussion and take help of mentioned articles below for discussion as well as introduction. Below are some references (titles of papers) which need to be cited in your manuscript. Try to extract as much information for interpreting your results from these papers as possible. 

Soil aggregation and soil aggregate stability regulate organic carbon and nitrogen storage in a red soil of southern China

Long-term fertilization enhanced carbon mineralization and maize biomass through physical protection of organic carbon in fractions under continuous maize cropping

Long-term fertilization alters chemical composition and stability of aggregate-associated organic carbon in a Chinese red soil: Evidence from aggregate fractionation, C …

Food and agricultural wastes-derived biochars in combination with mineral fertilizer as sustainable soil amendments to enhance soil microbiological activity, nutrient cycling …

Comparison of the Responses of Soil Enzymes, Microbial Respiration and Plant Growth Characteristics under the Application of Agricultural and Food Waste-Derived Biochars

The mechanisms of manure and biochar are widely discussed and reported in these suggestions which might help you make discussion more better. 

Another question is the application of N fertilizer at a rate of 270 kG per ha. What is the recommended rate at Jiangsu? It is more than the recommended rate I believe. Then what is the justification of applying that much?

Author Response

Dear Reviewer:

Thank you for reviewing our manuscript, and your comment is very professional and constructive. Based on your comment, we have for the second time made modifications to the original manuscript. Here, we responded to your good comments point by point, as below:

Round 3

Reviewer 1 Report

Comments and Suggestions for Authors

The authors have revised the manuscript according to suggestions. The work has been greatly improved and can be accepted for publication.